# SciBet as a portable and fast single cell type identifier

Chenwei Li[1,2,5], Baolin Liu[1,3,5], Boxi Kang [1,2,3], Zedao Liu[1,2], Yedan Liu[1,3], Changya Chen[4], Xianwen Ren[1,3✉] & Zemin Zhang [1,2,3✉]

Fast, robust and technology-independent computational methods are needed for supervised cell type annotation of single-cell RNA sequencing data. We present SciBet, a supervised cell type identifier that accurately predicts cell identity for newly sequenced cells with order-of-magnitude speed advantage. We enable web client deployment of SciBet for rapid local computation without uploading local data to the server. Facing the exponential growth in the size of single cell RNA datasets, this user-friendly and cross-platform tool can be widely useful for single cell type identification.

[1] Peking-Tsinghua Center for Life Sciences, BIOPIC and School of Life Sciences, Peking University, Beijing, China. [2] Analytical Biosciences Limited, Beijing, China. [3] Beijing Advanced Innovation Centre for Genomics, Peking University, Beijing, China. [4] Division of Oncology and Center for Childhood Cancer Research, Children's Hospital of Philadelphia, Philadelphia, PA 19104, USA. [5] These authors contributed equally: Chenwei Li, Baolin Liu. ✉email: renxwise@gmail.com; zemin@pku.edu.cn

The past decade has witnessed an exponential increase in the size of the dataset of single-cell RNA sequencing (scRNA-seq), as the cost per cell continues to decrease. For example, Tabula Muris[1] comprises more than 100,000 cells from 20 organs and tissues. MOCA[2], mouse organogenesis cell atlas, characterizes the single-cell map of mouse whole embryos with a newly developed technique sci-seq-3, which can measure millions of cells at a time. In addition, the Human Cell Project[3] (HCA) aims to characterize the single-cell map of all human cells, and its order of magnitude will reach billions. Facing such explosive data growth, one major challenge is the reliable and rapid cell type identification given a newly sequenced cell. Supervised cell type annotation of newly-generated data using annotated labels has become more desirable than unsupervised approaches, as unsupervised approaches tend to be far more laborious and computationally intensive. Traditional classification methods such as random forest classifier[4] (RF) and support vector machine[5] (SVM) are often time-consuming[6], whereas tools specifically designed for such tasks trade accuracy for speed[6] and integration-oriented tools[7] rely on computation-intensive search of anchor cells. Such practice can become inefficient if aforementioned huge datasets are used as reference.

Here we present SciBet to overcome these challenges. We use a multinomial-distribution model and maximum likelihood estimation to develop SciBet for accurate, fast, and robust single cell identification. Using a wide range of scRNA-seq datasets, including data from different biological systems and sequencing technologies, we demonstrate that SciBet outperforms other methods in accuracy, and especially in speed by a wide margin. In addition, SciBet balances high accuracy and low false positive rate by setting a null dataset as the alternate reference for cells with types not yet covered by the existing data. Finally, we provide both local and web-based SciBet implementations that are compatible with either existing or custom datasets for ultra-fast and accurate cell type identification.

## Results

### Overview of the algorithm.
The SciBet algorithm consists of 4 steps: preprocessing, feature selection, model training and cell type assignment (Fig. 1a–d, respectively). For a training dataset of scRNA-seq, we obtained the normalized expression matrix with common preprocessing pipelines (Methods) and calculated the mean expression values across cells with identical cell types, which were needed by the following steps (Fig. 1a). Because not all genes were equally useful for such the classification problem[6,8], we developed E-test to select the cell type-specific genes from the training set in a supervised and parametric manner, in order to remove the noisy genes as well as to accelerate the downstream classification by compressing the model. We first applied the statistic entropy in information theory to measure the dispersion degree for the Poisson-Gamma-mixture distributed gene expression, and the entropy could be directly estimated by the logarithm of the mean gene expression (Methods). We proposed the null hypothesis where all cell types were assumed not to be distinct and thus had the same mean and entropy. We then proposed a statistic $\Delta S$ as the total entropy difference, to measure the deviation of the observed mean expression from the mean expression under the null hypothesis. Under the criteria of feature selection by E-test, genes with larger $\Delta S$ tended to be more cell type-specific and would be kept by E-test for the downstream model training (Fig. 1b). After modeling the expression for each gene, we then modeled the expression across different genes by the assumption that the expression abundance of different genes was multinomially distributed in a given cell type (Methods). The parameters of each gene in the multinomial model could be directly estimated by the aforementioned mean gene expression after normalization in each cell type. These normalized parameters also represented the expression probability of each gene in a given cell type (Fig. 1c and Methods). We built multinormial models for each cell type in the training set, which composed the trained model of SciBet. For an unknown cell to be annotated by SciBet, we used its expression profile of the informative genes, and calculated the likelihood function over all multinomial models. SciBet selects the cell type whose model achieves the highest likelihood/prediction power in describing the distribution of the RNA profile. (Fig. 1d and Methods). Each cell in the test set was independently annotated.

### Performance assessment by cross-validation.
To perform quantitative benchmarks for such a multi-label classification problem, we applied the cross-validation strategy[9] as following: For each of the 14 datasets across multiple sequencing platforms (Supplementary Table 1), we trained a classifier with the randomly selected 70% of the cells (training set) and predicted the cell type for the remaining cells (test set), and repeated this entire procedure for 50 times. We applied the accuracy score[9], the ratio between the total number of correctly predicted cells against the number of all cells in the test set, as the performance metric in such cross-validation tasks (Methods and Supplementary Note 1). In the main figures, we calculated the mean accuracy across the 50 times repeats to represent the performance for each dataset.

To illustrate the performance and the scalability of the feature selection methods, we benchmarked E-test against F-test (one-way ANOVA) and M3Drop[8] using the same classifier scmap. Our results showed that E-test consistently achieved the highest classification accuracy. The superiority of E-test was independent on the number of selected genes and classification algorithms, indicating the robustness of E-test for identifying cell type-specific genes (Fig. 2a for all datasets together and Supplementary Fig. 1 of each dataset separately). In principle, E-test only needs linear operations and the computational time increases linearly with the numbers of cells and genes, in contrast to the quadratic increase for other gene selection methods. The time consumption and cell number relationship (Fig. 2b) confirmed this point, demonstrating the scalability of E-test for large datasets.

We benchmarked SciBet in terms of accuracy and speed, against scmap[6] and Seurat v3[7], with the aforementioned cross-validation strategy and E-test as the feature selection method. We found that SciBet achieved the best performance in most of the 14 benchmarking datasets, compared to the other methods (Fig. 2c for all datasets together and Supplementary Fig. 2 of each dataset separately). Considering the significance of the ability to detect rare cells, for each dataset, we measured the balanced accuracy score[9] (Methods and Supplementary Note 1) and the results showed that SciBet also achieved the best performance (Fig. 2d for all datasets together and Supplementary Fig. 3 of each dataset separately), indicating its superior ability to handle imbalanced datasets as well as to uncover rare cell types in a supervised manner. Finally, we evaluated how these classifiers behaved in speed using 4 simulated datasets comprising 500 genes along with 1000, 10,000, 20,000 and 50,000 cells, respectively. Our results demonstrated that SciBet out-performed scmap and Seurat v3 by orders of magnitude, with a classification speed of ~100,000 cells per second (Fig. 2e). We tested the performance of SciBet on two datasets, with both even and uneven cell type distributions. For a dataset of fresh peripheral blood mononuclear cells[10] after random down-sampling (Methods), where all cell types have equal proportions, the confusion matrices of classification showed that SciBet performed well on each cell type as opposed to scmap

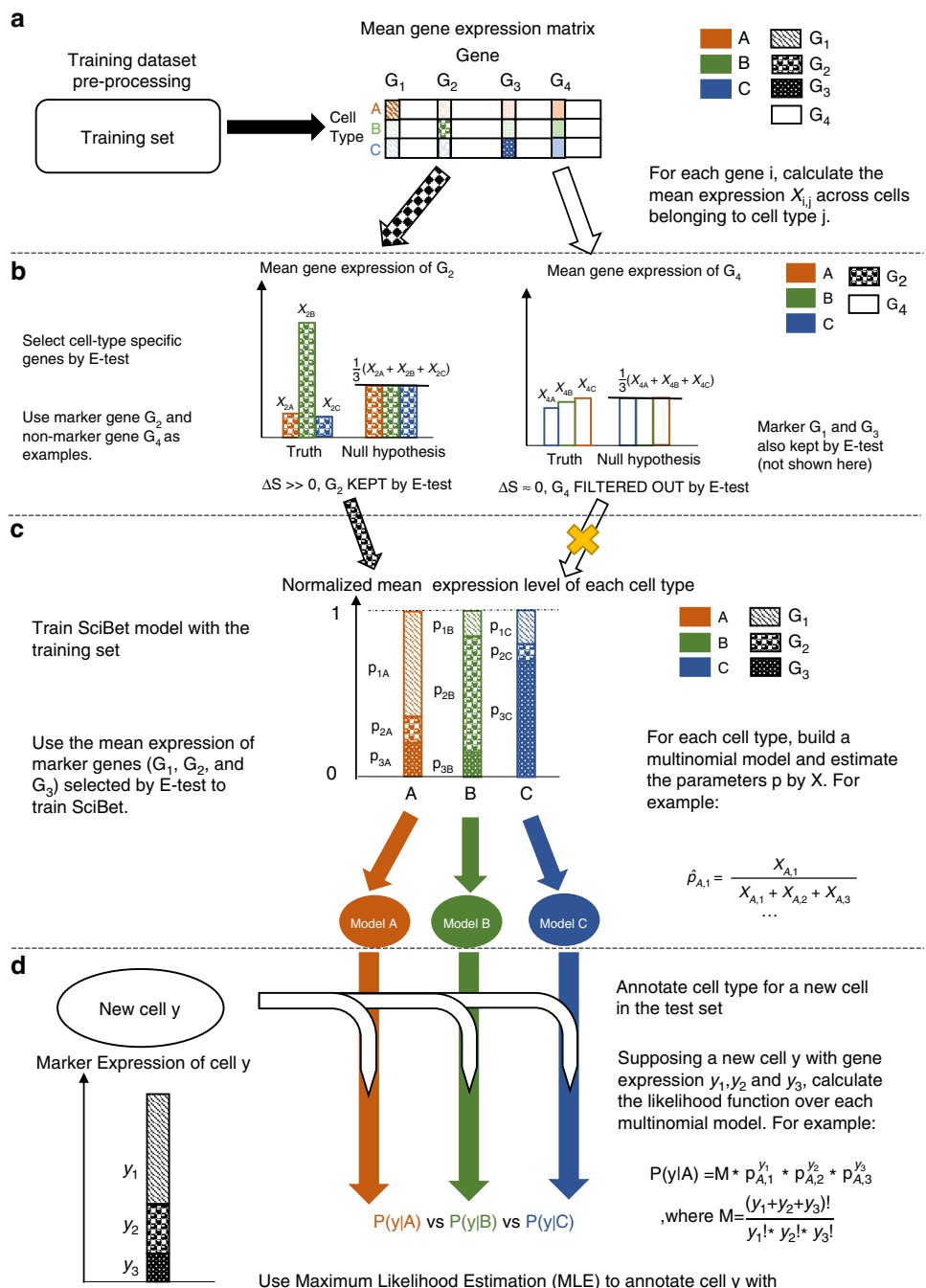

**Fig. 1 Overview of SciBet algorithm. a** Training set Pre-process by calculating the mean gene expression form the original expression matrix. Here we use marker genes $G_1$, $G_2$, and $G_3$ along with a non-marker gene $G_4$ as examples. **b** Using E-test to select cell type-specific genes for the downstream classification. Genes with total entropy difference larger than the predefined threshold will be kept. Genes selected by E-test are used for the model training and prediction. **c** Training SciBet model by obtaining the parameters for the multinomial models of each cell type. For each cell type, the sum of all parameters belonging to different genes equals to 1, which represent the expression probability of different genes. **d** Calculating the likelihood function of a test cell using the trained SciBet model and annotating cell type for the test cell with maximum likelihood estimation. Each cell in the test set is independently annotated.

and Seurat v3, which could not effectively discriminate close subtypes, such as CD4 memory T cells and CD4 regulatory T cells (Fig. 2f). In addition, for a human pancreatic dataset[11] with imbalanced proportions of cell types, SciBet could still provide the most discriminating power. By contrast, both Seurat v3 and scmap had lower accuracies in classifying delta cells, which accounted for only 3.8% of cells (Fig. 2g).

**Real-world applications of SciBet.** We benchmarked SciBet for cross-dataset annotation using one or more scRNA-seq datasets as training sets to predict the cell type in the test set. We first produced 6 instances of training-test pairs using four well-characterized human pancreas datasets[11–14] generated by distinct sequencing techniques, and predicted the cell types for one dataset with another dataset as training set (Supplementary

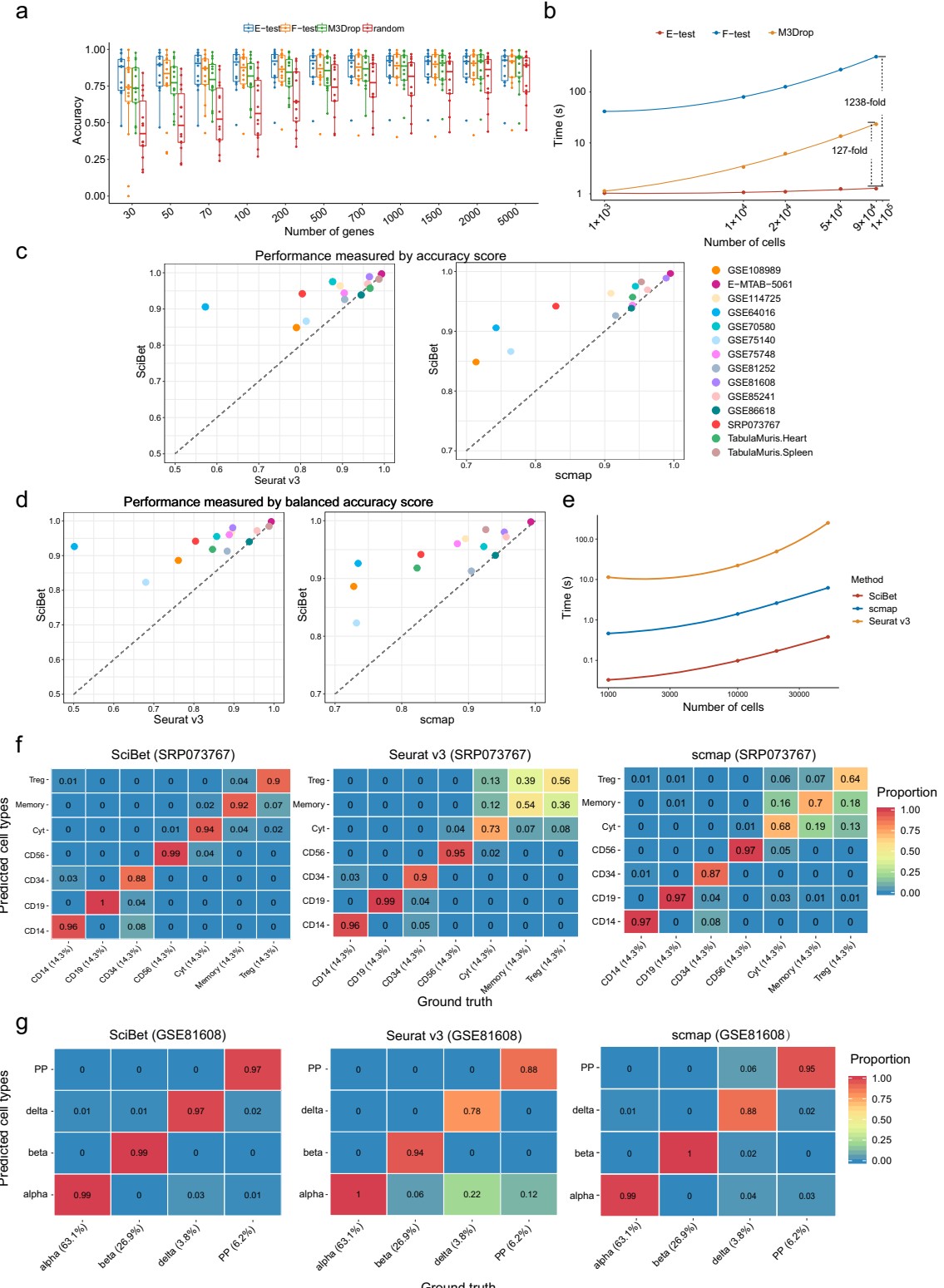

**Fig. 2 Cross-validation benchmarks. a** Performance of the feature selection methods measured by the accuracy score for $n = 14$ datasets (each dataset is plotted as an individual point, representing the mean accuracy score across 50 random repeats). Box plot shows the center line for the median, hinges for the interquartile range and whiskers for 1.5 times the interquartile range. **b** Single CPU consuming times for gene selection process with E-test, F-test and M3Drop (log scale). Solid lines are loess regression fitting (span = 2), implemented with R function geom_smooth. **c** Performance of the classifiers measured by the accuracy score for $n = 14$ datasets (each point represents the mean score across 50 repeats). **d** Performance of the classifiers is measured by the balanced accuracy score for $n = 14$ datasets (each point represents the mean score across 50 repeats). **e** Single CPU consuming times for classification (log scale). Solid lines are loess regression fitting (span = 2), implemented with R function geom_smooth. **f** Heatmap for the confusion matrix of the cross-validation result on the human PBMC dataset[10], with normalization for each column (origin label). **g** Heatmap for the confusion matrix of the cross-validation result on the human pancreatic dataset[11], with normalization for each column (origin label).

Table 2). For this test, SciBet showed a slight edge over Seurat v3 and both consistently outperformed scmap (Fig. 3a), indicating that the ultra-fast SciBet showed relatively equivalent performance in such a cross-platform classification task. As there were published mouse atlas datasets such as Tabula Muris while Human Cell Atlas was still in progress, we then tested whether SciBet could be used for cross-species classification, by training SciBet model with the Tabula Muris dataset and predicting cells types for human pancreas datasets. Only genes with one-to-one orthologues cross-species were used for feature selection and cell identification (Methods). Most cells (~92%) could be correctly mapped based on SciBet (Fig. 3b), demonstrating the utility of SciBet in cross-species cell identification.

We further collected 42 published human scRNA-seq datasets (listed in Supplementary Table 3) and built an integrated dataset to include the well-characterized and annotated major human cell types (Methods), which could be separated clearly by cross-validation with SciBet (Fig. 3c). This dataset, analogous to Tabula Muris[9], could serve as a plausible "mock" human cell atlas. We annotated the cell type for a recent human liver cell 10x genomics dataset[15] with the integrated data as reference. Sankey diagram revealed that major cell types were correctly predicted, including hepatocytes, endothelial cells and Kupffer cells (specialized macrophages located in the liver). In addition, closely related cell types such as NK and T cells, as well as B and Plasma B cells, could also be precisely discriminated by SciBet, suggesting the sensitivity and precision of SciBet for cross-platform prediction (Fig. 3d). Furthermore, each cell type could be further classified into more precise labels based on datasets uncovering novel sub cell types.

Due to the incomplete nature of reference scRNA-seq data collection, cell types excluded from the reference dataset may be falsely predicted to be a known cell type. Here we applied a null dataset as background, which is generated by mixing together all cell types in the datasets listed in Supplementary Table 4. For each cell in the test set, we quantified the likelihood to the reference set against that to the null set. Cells with smaller classification confidence score would be assigned as unassigned cells and thus be excluded from the downstream classification (Methods). As an example, we first considered a recent melanoma dataset[16] that comprises diverse cell types in the tumor microenvironment. We randomly sampled 70% of immune cells from the original data as reference, and regarded the remaining 30% immune cells and non-immune cells as query dataset, with cell types defined by the original authors. Proper classification of such query dataset should result in high classification confidence scores for immune cells while low scores for non-immune cells. We showed that SciBet consistently provided the best performance by achieving low false positive rate (FPR, the ratio of the number of falsely assigned cells against the number of cells that should be excluded) as well as high accuracy for the cells that needed to be assigned (Fig. 3e). Although Seurat v3 also properly controlled false positives, most NK cells were incorrectly assigned to CD8 T cells or the unassigned group, leading to false negatives. In addition, we also generated another 10 training-test pairs in a similar fashion using 16 datasets (Supplementary Table 5 and Methods), and demonstrated that SciBet correctly categorized >90% cells when FPR ranges from 0.001 to 0.05. Both scmap and Seurat v3 had a much lower accuracies with the same FPR cutoffs, indicating the superiority of SciBet in prediction accuracy and false positive control (Fig. 3f and Supplementary Fig. 4). In conclusion, SciBet controlled the potential false positives while maintaining high prediction accuracy for cells with types covered by the reference dataset.

We also noted that the features selected by E-test tended to be biologically meaningful genes. This can be seen by using nine immune cell types from 7 studies to test the performance of E-test for supervised gene selection (Supplementary Table 6). The top 54 genes with maximal $\Delta S$ were all well-established immune cell markers, known to play pivotal roles in corresponding cell types[17] (Fig. 3g). The identified marker genes also allowed interpretable visualization, with distinct immune cell population across different studies separately located in the 2D UMAP plot[18], further supporting their biological relevance (Fig. 3h).

**Web-based implementation of SciBet.** Based on ~100 well-annotated scRNA datasets collected from public repositories such as Broad Single Cell Portal, EMBL-EBI Single Cell Expression Atlas, NCBI Gene Expression Omnibus and CellBlast[19], we used SciBet to generate trained models for each dataset. The light-weight nature of trained models enabled the easy download together with the local SciBet package. For example, the size of a model with 100 cell types and 1000 signature genes would be no more than 1 MB. We further built a JavaScript version of SciBet (http://scibet.cancer-pku.cn), which bypasses the process of file uploading to a remote server. Users could use our web server to upload custom reference or test data for cell type prediction. For large query dataset that would take a long time for data transmission, we also provided a lightweight standalone package for local construction of the web-based tools by a simple command. This way, data files would be read and processed locally and directly in the web browser, with only small-sized models transferred from the server to the browser, thus achieving unprecedented speed and convenience.

## Discussion

SciBet addresses an important need in the rapidly evolving field of single-cell transcriptomics, i.e., to accurately and rapidly capture main features of diverse datasets regardless of technical factors or batch effect. Based on multiple benchmarks, SciBet achieves high prediction accuracy, while keeping low false positive rate for cells not represented previously. This advantage is achieved by considering not only the relative similarity to each cell type within the reference set, which is also used by scmap and Seurat v3, but also the absolute similarity to the entire reference set and null dataset (Methods). Both E-test and SciBet utilizes the fundamental concept that each cell type is represented by the simple mean expression vector. Thus, both methods only carry out a small number of linear operations on the expression matrix, and the entire computational process is efficient and can be thereby applied to very large-scale single cell studies emerging in coming years.

## Methods

**Data collection and pre-processing.** All scRNA-seq datasets in this paper were obtained from their public accessions. And we used the original cell type annotation provided by each publication as ground truth. For all datasets, we applied the common normalization methods as following. For read count data generated by full-length sequencing technique, we calculated Transcript Per Million (TPM)[20], added pseudo value one to handle 0 values and performed log-normalization. For unique molecular identifier (UMI) data, we applied the widely-used pre-processing methods proposed by Seurat v3[7] with default parameters (normalizing the UMI count of each cell with size-factor 10,000, adding one and then log normalization).

**Training-test split and cross-validation.** We implemented the hold-out validation strategy proposed by the python package sklearn[9] (function model_selection. train_test_split) as following: for each dataset, we applied the stratified random sampling without replacement to form the training and test set with the ratio 7:3, and repeated this process 50 times to obtain 50 training-test instances. Both the following feature selection and model training process were completely independent with the test set to avoid over-fitting, and the hold-out validation was performed for each dataset separately.

We implemented the accuracy score[9] (function metrics.accuracy_score in package sklearn) as the default performance measurement, which equals the ration between the total number of correctly assigned cells and the total number of all

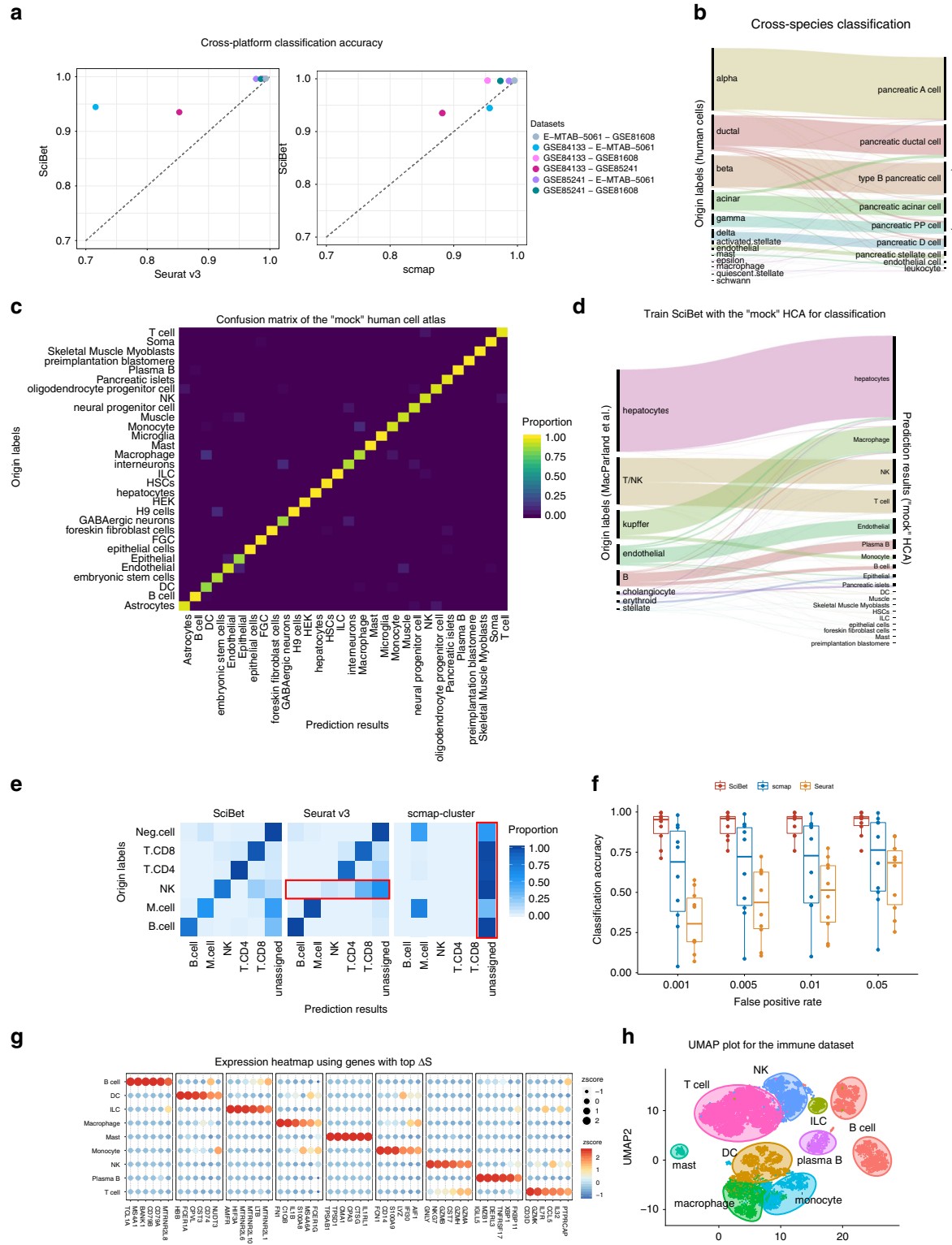

**Fig. 3 Applications of SciBet. a** Mean accuracy (across 50 repeats) for $n = 6$ cross-platform dataset pairs listed in Supplementary Table 2. **b** Cross-species classification with three human pancreas datasets projected to Tabula Muris dataset (Sankey diagram). The height of each linkage line reflects the number of cells. **c** Confusion matrix of the cross-validation result for 30 cell types in the "mock" human cell atlas (listed in Supplementary Table 3). **d** Single cell classification for a human liver dataset with integrated human dataset as reference, implemented by SciBet. **e** Confusion matrix for the case study of false positive control, with normalization for each row (origin label). Negative cells including malignant cells, CAF cells and endothelial cells were removed from the training set. Query cells with lowest classification confidence scores were labeled as unassigned. **f** False positive control evaluation with cell types not present in reference as negative cells, with $n = 10$ pairs of datasets (each point represents the mean accuracy score or FPR across 50 repeats). Box plot shows the center line for the median, hinges for the interquartile range and whiskers for 1.5 times the interquartile range. **g** Expression heatmap of the top 54 genes selected by E-test for the integrated immune dataset (Supplementary Table 6). **h** 2D-UMAP showing the dimensional reduction result based on the genes in **g**.

tested cells. We also implemented the balanced accuracy score[9] (function metrics. balanced_accuracy_score in package sklearn), which can be calculated as following: for each cell type in the test set, we calculate the correctly assigned ratio (recall), and then obtain the unweighted mean of recalls over each cell type (See detailed discussion in Supplementary Note 1).

**Supervised feature selection by E-test.** We used the statistic differential entropy from information theory ($S_{ij}$) to measure the dispersion degree of the expression distribution of a gene i in a given cell type j, and we could directly calculate this statistic by the equation:

$$S_{ij} = lnX_{ij} + h_i \qquad (1)$$

where $X_{ij}$ is the mean expression of gene i (ranges from 1 to m) across all cells belonging to cell type j (ranges from 1 to n) and $h_i$ is a gene-specific constant independent of cell groups, under the assumption that the gene expression is distributed as a Poisson-Gamma mixture[21] (See detailed derivation in Supplementary Note 2).

We then developed E-test to measure expression differences among n cell groups in the training set. First, we defined the null group 0, by averaging all pre-defined groups, and thus the null group had the mean gene expression of gene i as $X_{i0} = \frac{1}{n} * \sum_{j=1}^{n} X_{ij}$, the logarithm of the group-level arithmetic mean (AM) of $X_{ij}$. Under the null hypothesis that all cells from the pre-defined groups are randomly sampled from the the same cell population, the entropy for the null group 0 can be calculated as

$$S_{i0} = lnX_{i0} + a_i. \qquad (2)$$

Then we quantified the difference of gene i expression among the pre-defined groups by a static $\Delta S_i$, the total entropy difference of gene i, which equaled the total difference of the entropy from the null group 0 to the entropy for each pre-defined group j. Based on Eqs. (1) and (2), $\Delta S_i$ can be calculated as:

$$\Delta S_i = \sum_{j=1}^{n} \left( S_{i0} - S_{ij} \right) = \sum_{j=1}^{n} (lnX_{i0} + h_i - lnX_{ij} - h_i) \qquad (3)$$

Notably, Eq. (3) can be further derived as following:

$\Delta S_i = \sum_{j=1}^{n} (lnX_{i0} - lnX_{ij}) = n*\left( lnX_{i0} - \frac{1}{n} * ln \prod_{j=1}^{n} X_{ij} \right)$. Here we noticed that the second term $\frac{1}{n} * ln \prod_{j=1}^{n} X_{ij} = ln\left( \prod_{j=1}^{n} X_{ij} \right)^{\frac{1}{n}}$ served as the logarithm of the group-level geometric mean (GM) of $X_{ij}$. Then $\Delta S$ can be further written into a more intuitive form: $\Delta S_i = n*(ln \, AM_i - ln \, GM_i) = n* ln\frac{AM_i}{GM_i}$. According to Jensen's Inequality, $AM \geq GM$, and thus the $\Delta S$ will always be a non-negative number for each gene i. If $X_{ij}$ was a constant for each j, $GM_i$ would be equal to $AM_i$, and thus $\Delta S_i$ would equal 0, where the null hypothesis was satisfied.

Based on our assumption that $\Delta S$ of genes varies according to the nonuniformity of the expression among pre-defined cell groups, genes with larger $\Delta S$ tended to be more informative. Thus, for the training set we ordered genes based on their $\Delta S$ and selected genes with top $\Delta S$ for the downstream classification. The significance of $\Delta S$ for each gene can be obtained by the random permutation test (randomly permutate the cell group labels, calculate $\Delta S$ for each permutation and find the percentile of the actual $\Delta S$), which can be further accelerated by our strategy for approximation listed in Supplementary Note 2.4.

The entire feature selection process can be simplified and concluded as the following: for each gene, we calculated the mean expression of gene i across all cells belonging to cell type j, and then calculated $\Delta S_i$ as the log ratio between their group-level geometric mean ($\prod_{j=1}^{n} \bar{X}_i)^{\frac{1}{n}}$ and arithmetic mean $\frac{1}{n} * \sum_{j=1}^{n} \bar{X}_i$ across n cell groups. After obtaining $\Delta S_i$ for each gene i, we selected genes with top $\Delta S$ as informative genes (500 by default).

**Supervised cell type annotation by SciBet.** We applied the multinomial distribution to model the expression across different genes in a given cell type j. We assumed that mRNA molecules from the identical gene are equivalent and that the production for each mRNA molecule is independent. Thus, for a type-j cell, we could denote the probability of producing a mRNA molecule belonging to gene i as $p_{ij}$, where i ranged from 1 to n. Given the cell type j, the vector $\mathbf{p}_j = [p_{1j}, .., p_{ij}, .., p_{mj}]$ across all m informative genes would serve as the parameters of a multinomial distribution, where $\sum_i p_{ij} = 1$. For a cell y belonging to cell type j, the probability of having the expression profile $\mathbf{y} = [y_1,..,y_i,..,y_n]$ can be calculated as $P(y|j) = \frac{(\sum_i y_i)!}{\prod_i (y_i!)} \prod_i (p_{ij}^{y_i})$.

After processing the training dataset with aforementioned steps including scaling, log normalization and feature selection, we calculated $X_{ij}$ as mentioned in E-test section, the mean expression of gene i across all cells belonging to cell type j. For each cell type j, $p_{ij}$ across all gene i could be estimated by $X_{ij}$ after normalization across genes and Laplace Smoothing (adding one to account for zero values), which can be written as $\hat{p}_{ij} = \frac{1 + X_{ij}}{\sum_i (1 + X_{ij})}$.

According to the multinomial assumption, the likelihood function of an unknown cell y in the test set with gene expressions profile $\mathbf{y} = [y_1,..y_i,..,y_n]$

belonging to type j can be written as $P(y|j) = \frac{(\sum_i y_i)!}{\prod_i (y_i!)} \prod_i (p_{ij}^{y_i})$. We assign cell type j to cell y with maximum likelihood estimation (MLE) as following:

$$\hat{j} = argmax_j(P(y|j)) = argmax_j \left( \frac{(\sum_i y_i)!}{\prod_i (y_i!)} \prod_i \left( p_{ij}^{y_i} \right) \right) = argmax_j \left( \prod_i \left( p_{ij}^{y_i} \right) \right) \qquad (4)$$

Equation (4) can be further derived into $\hat{j} = argmax_j(\prod_i (p_{ij}^{y_i})) = argmax_j(\sum_i y_i * lnp_{ij}) = argmax_j(\sum_i (q_i * lnp_{ij} - q_i * lnq_i)) = argmin_j D_{KL}(q||p_j)$, where $q_i = y_i / \sum_i y_i$. It indicates that we annotate the test cell to such the cell type that its mean expression vector of all marker genes has the minimal Kullback–Leibler divergence to the unknown cell y.

The discussion about using the cell type prior probability and thus using Bayes decision can be seen in Supplementary Note 3.

**Seurat v3 and scmap.** For cross-validation benchmarks, we performed the classification using the function scmapCluster of scmap without considering the unassigned cells. For Seurat v3, we identified anchors and classified cells using the FindTransferAnchors function and TransferData functions, respectively, with the default parameters.

**PBMC dataset with even proportions of cell types.** We implemented the down sampling strategy proposed by python package imbalanced-learn[22] (under_-sampling.RandomUnderSampler) as following: we randomly sampled all 7 cell types to a fixed number 2500, which was smaller than the size of all cell types. Then we performed the training-test split and hold-out validation strategy as mentioned before.

**"Mock" human cell atlas.** We integrated 42 human datasets covering the major cell types, which means the canonical and well-characterized cell types previously identified without single cell RNA sequencing data and can be usually mapped to the cell type knowledgebase (e.g., EBI Cell Ontology). For 26 of the 42 datasets, we downloaded raw scRNA-seq fastq files, and estimated the gene-level expression abundance with kallisto[23] and human genome reference hg19 (downloaded from UCSC).

We curated the cell label by unifying the major cell type annotated by different publications (e.g., considering both "T lymphoid cells" and "T cells" as "T cells"). We neglected the minor cell types, which means those novel subtypes uncovered by scRNA-seq with unsupervised clustering and annotation (e.g., considering both "DC_cluster1" and "DC_cluster2" as DC cells).

**False positive control for cell type assignment.** The basic idea is to quantify the relative likelihood of each cell to the specified training dataset against that to the collection of the entire datasets, which serves as the background. To control potential false positive predictions, we first identified common marker genes (500 genes by default) exclusively expressed in reference cells using E-test, with a null dataset (by integrating all datasets listed in Supplementary Table 4) as the background. Then we considered the specified training set and the null set as two pseudo cell types, and defined the classification confidence score C of each query cell, by calculating the ratio of the likelihood functions based on the multinomial assumption mentioned before, as following:

$$C = \frac{\prod_i p_{ir}^{y_i}}{\prod_i p_{in}^{y_i}} \qquad (5)$$

In Eq. (5), $y_i$ is the expression value of gene i, and $p_{ir}$ and $p_{in}$ are expression probability of gene i estimated by considering the entire training dataset and the entire null dataset as two pseudo cell types, respectively. For false positive control benchmarking experiments, we did not assign cell types to the cells with lowest score C with a series of cutoffs from 0.001 to 0.5.

We generated 10 training-test pairs using 16 datasets listed in Supplementary Table 5. Here we take training-test pair 1 (generated by GSE104276[24] and GSE108989[25]) as an example. For dataset GSE104276, which comprised cells from human prefrontal cortex, we randomly sampled 70% of the cells as reference. Cells from GSE108989 (T cells) are not represented in the reference, and thus could be considered as negative cells. The remaining 30% cells of GSE104276 and cells from GSE108989 together formed the query dataset. For each training-test pair, we repeated the entire procedure including random sampling and classification for 50 times.

**Cross species classification.** We used the HomoloGene databases provided by NCBI (Build 68) to identify homologous genes between human and mouse, and kept only genes that have one-to-one correspondence, which serves as a look-up table. After obtaining the intersection gene set between the training dataset (Tabula Muris) and the look-up table, the gene names were then converted to human gene names to obtain a test-set-compatible training set for classification.

**Hardware platform**. All benchmarks were performed on a computer with AMD Threadripper X2990, 128GB DDR4 memory and Seagate 2TB HDD.

**Reporting summary**. Further information on research design is available in the Nature Research Reporting Summary linked to this article.

## Data availability

All single cell gene expression datasets that support the findings in this study were obtained from their public accessions. The detailed information including the accession codes and publication citations for all datasets can be seen in Supplementary Information.

## Code availability

All the functions mentioned above were implemented in the R package SciBet, which can be downloaded at http://scibet.cancer-pku.cn. An online version of SciBet is also available at this website, which is based on JavaScript. All codes used for benchmarks are available at https://github.com/PaulingLiu/scibet.

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

## Acknowledgements

This project was supported by Beijing Advanced Innovation Centre for Genomics at Peking University, National Natural Science Foundation of China (31530036, 91742203 and 91942307). We thank the Computing Platform of the CLS (Peking University) for providing computing resource.

## Author contributions

Z.Z. conceived this study. C.L. developed the algorithm of E-test and SciBet; B.L. performed the benchmark analyses. C.L. and Z.L. implemented the JavaScript version of SciBet. B.L. and B.K. developed the R package. Z.L., C.C. and Y.L. produced and deployed the trained models on the web server. C.L., B.L., X.R. and Z.Z. wrote the manuscript with all the authors' inputs.

## Competing interests

C.L., B.K., Z.L. and Z.Z. are either interns, employee, or founder of Analytical Biosciences Limited. The remaining authors declare no competing interests.
