## [Peer Review File · Nature Communications]

Reviewers' comments:

Reviewer #1 (Remarks to the Author):

In this manuscript the authors propose a new method for cell type annotation of cells from single-cell RNAseq profiling. The context for the method is supervised learning, i.e. using labelled (cell types) training data to train (estimate) a classification model, which can subsequently be applied to predict class label (cell type) on new single-cell RNA sequencing data. The authors highlight computational cost being a limiting factor in previously established methods, and highlight computational efficiency/speed of the proposed method.

Unfortunately the manuscript lack clarity in description of the details of the method and algorithm, and the manuscript also struggle with a weak description of how theory underpins the methodology as well as some key assumptions made. Based on what can be understood, the proposed algorithm consist of several components:

- 1) identification of key genes for classification of cell type (variable selection) using an Entropy-based measure
- 2) A classifier (trained in a supervised fashion) that outputs class probabilities for cell types - used to assign cell type of new single-cell RNAseq data.

The authors attempt to demonstrate / validate the method on a number of publicly available data sets (cross-validation). However, the process of variable selection, training of the model, and subsequent empirical evaluation is not transparent and clear enough to allow assessment of exactly the parameters were estimated, which data were used, how variable selection was performed etc, etc.

The manuscript addresses a relevant problem, but lack of clarity in many respects makes it hard for the reader to both understand the method in detail, as well as determine the quality of the empirical support for the performance of the method.

Specific comments

- 1) It appears that the cell type data is fully confounded with different studies, this is a major issue, as it then is not possible to determine if the variable selection procedure or classification performance is driven mainly by study or actual cell type. It would be essential to have annotated (Cell type) data within the same study, using the same protocol, in the same lab, using the same SC sequencing method etc.
- 2) It is unclear what constituted the reference set for training
- 3) was the variable selection using the E-test only based on training data? or was it applied a prior step using all data, which would inevitably lead to over-fitting
- 4) Many abbreviations are not defined nor cited, including e.g. TPM and CPM.
- 5) L61. suddenly clustering is referred to without any explanation. What is the clustering procedure? What is the method, how is it applied here?
- 6) is the E-test as described in the suppl methods applied in a supervised method just to select variables? Since cell type appear to be confounded with study, you are likely to optimize for variables that are good to distinguish between different studies (cell lines), rather than actual cell type?
- 7) Fig 1 / L412, and also related section in main text: no detail around how the cross-validation test was performed. Was all parameters fitted and optimized only on training split data (including variable selection, and other optimization relevant aspects)? Appears to be a high risk of

overfitting, but the lack of detail make it impossible to assess.

8) Fig 2B, how were the 8 data sets selected out of the large number of data sets you have analysed?

9) Fig 2B. why do you pool results across multiple studies? Suppl table 3 is hard to localize due to cryptic file naming in the reviewer .zip bundle (probably due to submission system).

10) L233 "Pseudo TPM or CPM 1

has been added to all element to the expression matrix. ". Please clarify further, unclear what this means? is this simply to handle 0 counts?

11) L244. Either add citation, or provide derivation of the differential entropy for Gamma distribution

12) What is the justification for assuming that the shape param alpha constant across all genes? Do this relate to anything reasonable physical aspect of the SC expression data? You need to clarify this further and motivate.

12) L252, Unclear internal reference to the S-E formula " gene-independent constant ($\square - \square$ formula, Methods)*, hard to establish which formula this is (consider add numbers to definitions /equations)?

13) L273, "we observed that Delta(S)' followed a Normal distribution", please provide either theoretical justification, or comprehensive Empirical evidence for this claim. As it stands now, there is no support for this claim at all, and it is hard to be convinced that this is reasonable just based on this brief statement.

14) L305, Please elaborate further why you claim this is a Bayesian decision? Especially since you ignore prior on classes (which is likely to be of importance too!), it does not appear to be that Bayesian (application of Bayes rule does not necessarily imply that you take Bayesian approach, which generally would be consistent probabilistic reasoning, to establish posteriors based on prior and likelihood)?

15) Computational speed is of course valuable. However, the actual computational burden of e.g. fig 2c, is not problematic in almost any real-world situation. Accuracy is much more important, but in this respect, performance appear to be similar to Seurat?

Reviewer #2 (Remarks to the Author):

This manuscript presents SciBet, a model based supervised cell type identifier. SciBet trains a generative model, which contains an array of probabilities of drawing mRNAs from specific genes. SciBet uses multinomial distributions to model the generation of mRNAs. It assumes that single cell sequencing is a sampling with replacement process: mRNA is sampled from genes with gene-specific probabilities. In training, SciBet computes a set of gene probability profiles, one for each cell type. In testing, given a cell's mRNA profile, SciBet assigns the cell a type by computing a series of posterior probabilities. SciBet iterates through all trained cell types. For each cell type it computes the posterior probability of observing the mRNA profile of the test cell, assuming that the test cell is of the trained cell type. The cell type that generates the highest posterior probability is designated as the type of the test cell. The authors compared against scmap-cluster and Seurat v3 and showed that SciBet has a similar accuracy with Seurat v3 while being significantly faster. It also compared E-test against F-test, M3Drop and showed that E-test improves the accuracy of SciBet when selecting the top 1000-or-fewer genes. This tool is urgently needed in the single cell field because classic clustering algorithm does not annotate the cells into biological category. The web portal is user friendly. The idea of setting a null cell type is interesting. The efficiency of SciBet is impressive (~100x faster than Seurat v3).

I also have some concerns and comments below.

1. The paper did not mention how the general cell population is designed. Do all cell types have equal proportions in the general population? How are under-represented cell types handled? Author states "We further collected 42 published human scRNA-seq datasets with full length mRNA coverage and built an integrated dataset to include major human cell types that could be separated clearly by self-projection with SciBet". There is no information how these 42 reference datasets in the paper.

2. The E-test only shows slight improvements when the number of selected genes for training is small. When the number of genes is beyond 500, E-test have similar, and sometimes even inferior, performance than F-test. Judging from the accuracy plot (Figure 1b), SciBet achieves the highest accuracy with around 2k genes. However, with 2k genes, the performance of E-test shows little difference from F-test. It is not clear why E-test is preferred over F-test.

3. It is unclear how this web will be useful for user's customized datasets that are not presented in the reference datasets. It will be interesting to introduce how to train another model on own data in the tutorial.

4. The authors introduced E-test without providing a motivation. We suggest the authors to introduce SciBet and its generative multinomial-distribution based mRNA sampling model first. Then give a motivation of why feature selection is important, before introducing E-test. We also prefer the authors to support the motivation of E-test in the Result section, perhaps presenting the prediction accuracy of training SciBet without feature selection.

6. The mechanism of E-test is not clearly stated in the main text. We infer that E-test is trying to finding cell-type specific genes that differs from the total population. However, this idea was not clearly stated in the paper.

7. The current manuscript is a bit hard to follow. There are many typos, repeats and undefined concepts. We strongly recommend the authors to significantly improve the writing quality of the manuscript in their revision. In addition, more detailed information (e.g. number of cells, sequencing depth) about reference and test datasets need to be provided.

Minor comments:

9. The authors use the term "projection" as cell type annotation. We suggest the authors to avoid this term, as it infers that SciBet is projecting a testing dataset to a training dataset, which is not what SciBet does.

10. Line 88-96 is a repeat of the first 3 paragraphs.

11. It is not clear in the main text that SciBet needs to be trained with a curated dataset first. It is also not clear in the main text what is a null dataset and what is its purpose.

12. Line 75: delta S is not defined.

13. Line 147-148: Sentence "We projected a recent human liver cell 10x genomics dataset to the integrated data by SciBet." is unclear. Perhaps SciBet trained a model with an integrated data and the model was tested using a 10x liver dataset?

14. Line 154-156: Sentence "Furthermore, each major cell type could be further classified into the minor label based on its corresponding dataset." is not clear. What are major labels and minor labels? How are these labels curated? Are minor labels included in the training dataset or SciBet

infers minor labels in an unsupervised manner?

15. Line 32: unnecessary line break.

16. Some names of functions in this tool are ambiguous, like `C_heatmap()` and `N_heatmap()`.

17. The R documentations of many functions are over-simplified.

18. There are still some little mistakes in the tutorial, like the process of loading model.

Reviewers' comments:

Reviewer #1 (Remarks to the Author):

In this manuscript the authors propose a new method for cell type annotation of cells from single-cell RNAseq profiling. The context for the method is supervised learning, i.e. using labelled (cell types) training data to train (estimate) a classification model, which can subsequently be applied to predict class label (cell type) on new single-cell RNA sequencing data. The authors highlight computational cost being a limiting factor in previously established methods, and highlight computational efficiency/speed of the proposed method.

Unfortunately the manuscript lack clarity in description of the details of the method and algorithm, and the manuscript also struggle with a weak description of how theory underpins the methodology as well as some key assumptions made. Based on what can be understood, the proposed algorithm consist of several components:

- 1) identification of key genes for classification of cell type (variable selection) using an Entropy-based measure
- 2) A classifier (trained in a supervised fashion) that outputs class probabilities for cell types - used to assign cell type of new single-cell RNAseq data.

The authors attempt to demonstrate / validate the method on a number of publicly available data sets (cross-validation). However, the process of variable selection, training of the model, and subsequent empirical evaluation is not transparent and clear enough to allow assessment of exactly the parameters were estimated, which data were used, how variable selection was performed etc, etc.

The manuscript addresses a relevant problem, but lack of clarity in many respects makes it hard for the reader to both understand the method in detail, as well as determine the quality of the empirical support for the performance of the method.

We thank the reviewer for pointing out the major advantages of SciBet and also greatly appreciate all the constructive suggestions and criticisms, which have helped us to further improve the novelty and quality of our manuscript. Since the previous manuscript was directly transferred from another journal, which required a shorter format, the manuscript was constrained by a different word limit and style. Regarding the major concern of lacking clarity in description of the details of the method, algorithm and key assumption, here we point out the major improvements in our revised manuscript, especially after integrating all specific comments:

- i> We have improved our writing qualities to avoid ambiguous or misleading descriptions and added detailed information of how the algorithm works and benchmarks are performed, etc. We have released the codes for benchmarking on GitHub, (<https://github.com/PaulingLiu/scibet>), along with more detailed R documentation for the SciBet package.

- ii> We have given detailed derivation or added citations for the key assumptions, which were not well presented in the previous manuscript. In addition, we have now added new figures for a more intuitive comparison and visualization (e.g., presenting the classification results within each dataset separately).
- iii> We have further demonstrated the potential advantages of SciBet by multiple and real-world examples as case studies, especially regarding the speed and computational efficiency which will become necessary when tackling datasets with tens of millions of cells.

Specific comments

1) It appears that the cell type data is fully confounded with different studies, this is a major issue, as it then is not possible to determine if the variable selection procedure or classification performance is driven mainly by study or actual cell type. It would be essential to have annotated (Cell type) data within the same study, using the same protocol, in the same lab, using the same SC sequencing method etc.

We first apologize for the ambiguous description which led to misunderstandings, and would like to clarify that the results in both Fig. 1b and 2b were performed on each dataset separately, rather than integrating them together. What we did was to pool the accuracy of each dataset together to reflect the overall performance across datasets for visualization. We have now corrected this problem from 3 aspects:

- i> We have now updated the manuscript with more details about how the training and testing processes were made and how the benchmark was performed. Specific answers can be seen in our replies to comments 2, 6, 8 and 9.
- ii> For each dataset in the cross-validation benchmarks, we generated 50 instances of training-test subsets by random split (See details in the reply to comment 2). We have now added the benchmark results for each study separately (n=50) in the Supplementary Fig. 1-3.
- iii> For each of the 14 datasets (listed in Supplementary Table 1), we used the mean accuracy across the 50 instances to represent the performance in the main figures, in order to show the overall performance across all datasets.

2) It is unclear what constituted the reference set for training

In the revised version of manuscript, we have now added details about what constituted the reference set, from the following aspects:

i> Dataset pre-processing (in section METHODS):

“All scRNA-seq datasets in this paper were obtained from their public accessions (Supplementary Table 1-6). And we used the original cell type annotation provided by each publication as ground truth.”

ii> Train-test split and cross validation (in section RESULTS and METHODS)

“For each of the 14 datasets across multiple sequencing platforms (Supplementary Table 1), we trained the classifier scmap with the randomly selected 70% of the cells (training set) and predicted the cell type for the remaining cells (test set), and repeated this entire

procedure for 50 times.”

iii> Real-world applications of SciBet (in section RESULTS and METHODS)

For the cross-datasets performance assessment (“Application in real-world situations” part of the section RESULTS), we have now clarified which dataset served as training set, such as the cross-species benchmark with datasets listed in Supplementary Table 2.

3) was the variable selection using the E-test only based on training data? or was it applied a a prior step using all data, which would inevitably lead to over-fitting

Thanks for reminding us to clarify the details about the feature selection step. We confirm that the feature selection step is completely independent with the test set. We have clarified this point in the methods part and figure legends in the revised manuscript.

4) Many abbreviations are not defined nor cited, including e.g. TPM and CPM.

We apology for this oversight. We have corrected this problem in the revised manuscript. For example, we have added detailed explanations along with the citation for TPM, as following:

“For read count data generated by full-length sequencing technique, we calculated Transcript Per Million (TPM)¹, added pseudo value one to handle 0 values and performed log-normalization.”

Furthermore, we have carefully examined all abbreviations and citations in the revised manuscript, including MOCA (mouse organogenesis cell atlas), HCA (human cell atlas), RF (random forest), SVM (support vector machine) and FPR (false positive rate).

5) L61. suddenly clustering is referred to without any explanation. What is the clustering procedure? What is the method, how is it applied here?

The previous description was indeed misleading. Here the “clustering” was meant to refer to the cell type label or pre-defined cell groups, providing the group information for the supervised learning rather than the group information from a specific publication. We have now corrected the mistake in the revised version as following (can now be seen in “Overview of the algorithm” part of the section RESULTS):

“We developed E-test to select the cell type-specific genes in a supervised and parametric manner... We then proposed a new statistic ΔS , the total entropy difference to measure the deviation of the observed mean expression of cell groups against the null hypothesis, where all cell types were assumed not to be distinct and thus had the same mean and entropy.”

In addition, detailed derivations and explanations can be seen in “Supervised feature selection by E-test” part of the section METHODS, along with Supplementary Note 2.

6) is the E-test as described in the suppl methods applied in a supervised method just to select variables? Since cell type appear to be confounded with study, you are likely to optimize for variables that are good to distinguish between different

studies (cell lines), rather than actual cell type?

We would like to confirm that the results in previous Fig. 1b were performed on each dataset separately, and thus cell type prediction would not be confounded with studies. We have now added details about this point in the concept map (Fig 1) and main text (“Concept map of the algorithm” part of the section RESULTS). To clarify this further, we have now also added the performance benchmark of E-test within the same study separately in the Supplementary Fig 1.

7) Fig 1 / L412, and also related section in main text: no detail around how the cross-validation test was performed. Was all parameters fitted and optimized only on training split data (including variable selection, and other optimization relevant aspects)? Appears to be a high risk of overfitting, but the lack of detail make it impossible to assess.

First, we would like to confirm that the feature selection step and the training step are completely independent of the test set. To address this criticism, we have now clarified our strategy in “Performance assessment by cross-validation” part of the section RESULTS: *“For each of the 14 datasets across multiple sequencing platforms (Supplementary Table 1), we trained the classifier scmap with the randomly selected 70% of the cells (training set) and predicted the cell type for the remaining cells (test set), and repeated this entire procedure for 50 times.”* In addition, the procedure of training and test can be visualized in Fig 1.

8) Fig 2B, how were the 8 data sets selected out of the large number of data sets you have analysed?

We thank the reviewer for this helpful question. We randomly selected those 8 datasets without any preference. However, the question by this reviewer prompted us think that the selection of those 8 datasets might leave readers the impression that these were somehow handpicked. To avoid any confusion, here we added the remaining 6 datasets, so that the entire collection of 14 datasets (Supplementary Table 1) are included. The newly updated evaluation analysis, which are now included in several new figures (Fig. 2c-d, Supplementary Fig. 2-3), still supports our previous conclusion that SciBet achieves the highest accuracy of cross-validation tasks.

9) Fig 2B. why do you pool results across multiple studies? Suppl table 3 is hard to localize due to cryptic file naming in the reviewer .zip bundle (probably due to submission system).

Here we would like to explain that we pooled the results to present the overall performance across different datasets, and would like to clarify that in Fig. 1B in the previous paper was performed for each study separately. In our revised manuscript, to avoid confusion, we have modified the style of result representation in Fig. 2c-d (each point reflects the mean accuracy of cross-validation across 50 repeats for each dataset),

and have added the performance benchmark of SciBet within the same study separately in the Supplementary Fig 2-3.

We will send the editor another zip file containing all needed files to avoid the file naming problems.

10) L233 “Pseudo TPM or CPM 1 has been added to all element to the expression matrix. “. Please clarify further, unclear what this means? is this simply to handle 0 counts?”

The reviewer is correct for such inference. It is widely used in common analysis pipelines such as Scanpy² and Seurat³. To avoid confusion, we have now added details along with the answer for the third point in the “Data collection and pre-processing” part in section METHODS:

“For read count data generated by full-length sequencing technique, we calculated Transcript Per Million (TPM)¹, added pseudo value one to handle 0 values and performed log-normalization.”

11) L244. Either add citation, or provide derivation of the differential entropy for Gamma distribution

We have now added the citation⁴. To clarify this further, we have now moved the detailed derivation to Supplementary Note 2.1 in the revised version. In addition, we have refined the structure of the algorithm description in “Supervised feature selection by E-test” in section METHODS for a better logical flow, which can be summarized as following:

- i) Derivation for the differential entropy estimated by the mean gene expression under the Poisson-Gamma distribution assumption.
- ii) Definition of ΔS and derivation for the calculation of ΔS based on i).
- iii) An equivalent but more intuitive form of ΔS : the ratio between the group-level arithmetic mean and group-level geometric mean.
- iv) The criterion for select most informative genes.

12) What is the justification for assuming that the shape param alpha constant across all genes? Do this relate to anything reasonable physical aspect of the SC expression data? You need to clarify this further and motivate.

Here we have given the citation and explanation for the assumption. However, we have further realized that a weaker assumption would suffice, which is now in replacement of the previous assumption.

In the previous version, we added the constraint for the parameters of a given gamma distribution in order to reduce the complexity of the model (as α cannot be analytically estimated⁵) and avoid the risk of over-fitting. We have applied the assumption proposed by SAVER⁶ which modeled the expression data using the Poisson-Gamma mixture, where the parameter α could be always assumed as a constant across all genes. This was explained as that all genes have constant coefficient of variation (Section “Constant

Coefficient of Variation" in Supplementary Text of SAVER paper), as following:

Constant Coefficient of Variation

Under the constant coefficient of variation assumption, we assume that the variance scales quadratically with the mean. This corresponds to a typical constant scaling of a Gamma distribution, since scaling by a constant c still gives a Gamma distribution with mean scaled by c and variance scaled by c^2 . The coefficient of variation CV^2 can be expressed as

$$CV_g^2 = \frac{v_{gc}}{\mu_{gc}^2} = \frac{1}{\alpha_g}$$

Thus, assuming a constant coefficient of variation is equivalent to assuming a constant shape α_g parameter in the usual Gamma distribution parametrization. Therefore, we have the fol-

We have further realized that a weaker assumption would suffice, in which α is assumed as a gene-specific but not cell type-specific parameter, because the final expression for ΔS does not include a gene-specific α parameter:

$$\Delta S_i = \sum_{j=1}^n (S_{0j} - S_{ij}) = \sum_{j=1}^n (\ln X_{i0} + h_i - \ln X_{ij} - h_i) \quad (***)$$

And this can be explained as that for each specific gene, the expression fold change of the different cell type is caused by a linear scaling factor. The discussion above can also be seen in Supplementary Note 2.2.

12) L252, Unclear internal reference to the S-E formula " gene-independent constant (???? - ???? formula, Methods)*, hard to establish which formula this is (consider add numbers to definitions /equations)?

We apologize for this inconvenience and have now added markers to those definitions or equations to the "Supervised feature selection by E-test" part in METHODS.

13) L273, "we observed that Delta(S)' followed a Normal distribution", please provide either theoretical justification, or comprehensive Empirical evidence for this claim. As it stands now, there is no support for this claim at all, and it is hard to be convinced that this is reasonable just based on this brief statement.

We apologize for the seemingly sudden claim of normal distribution for $\log(\Delta S)$, and have replaced this with the following statement in "Supervised feature selection by E-test" part of METHODS, as following:

"The significance of ΔS for each gene can be obtained by the random permutation test (randomly permute the cell group labels, calculate ΔS for each permutation and find the percentile of the actual ΔS), which can be further accelerated by our strategy for approximation listed in Supplementary Note 2.4."

14) L305, Please elaborate further why you claim this is a Bayesian decision? Especially since you ignore prior on classes (which is likely to be of importance too!), it does not appear to be that Bayesian (application of Bayes rule does not necessarily imply that you take Bayesian approach, which generally would be consistent probabilistic reasoning, to establish posteriors based on prior and likelihood)?

We apologize for the unclear explanation. In the revised version, we replace the **Bayesian decision** with **Maximum Likelihood Estimation**, without considering the class prior probability. In addition, we have now added the following discussion on this issue in Supplementary Note 3.

“The proportion of each type of cell in the data measured by a single cell sequencing does not necessarily and correctly reflect the prior probability of appearance. For example, if a piece of tissue is sequenced without any sorting, then the proportion of each cell type in the results of single-cell sequencing can reflect the prior probability. However, if certain artificial filtering (such as Fluorescence-activated cell sorting to select cells highly expressing certain surface protein) is performed, or the dataset is integrated from different batches or studies, then the final cell type ratio at this time cannot correctly reflect the prior probability of the appearance of such cell types. This latter situation is more common and considered to be more appropriate for maximum likelihood estimation (i.e., the prior probabilities of each class are considered equal in Bayesian decision making), which is used as the default option.

*If users choose to consider the prior probabilities of different cell types, they can replace the strategy for making decision (formular (****) in part ‘Supervised cell type prediction by SciBet’, section METHODS) with the following strategy of Bayes Decision Rule $\hat{j} = \operatorname{argmax}_j (P(y|j) * P(j)) = \operatorname{argmax}_j (\prod_i (p_{ij}^{y_i}) * P(j))$, to make decisions. And in most cases, users can estimate the prior probability according to the proportion of cell types in the training set.”*

15) Computational speed is of course valuable. However, the actual computational burden of e.g. fig 2c, is not problematic in almost any real-world situation.

The reviewer is correct in pointing out that, in early years when the data volume is low, the computational speed might not be critical. However, the past decade has witnessed an exponential increase in the size of datasets of single-cell sequencing (shown by the following figure⁷), as the cost per cell continue to decrease. Tabula Muris⁸, which was used in our benchmarks, comprises more than 100,000 cells from 20 organs and tissues. Besides, MOCA (mouse organogenesis cell atlas)⁹ characterizes the single-cell map of mouse, which can sequence millions of cells at a time. The Human Cell Project (HCA)¹⁰ aims to characterize the single-cell map of all human cells, and its order of magnitude will reach billions. We chose these thousands of cell-level datasets for practical testing because other software is difficult to run directly on the above datasets in common servers without any down-sampling operation. Thus, SciBet not only shows great advantages

currently, but also will be more valuable in the future.

Editorial Note: Figure reproduced under an Open Access licence from Angerer, P. et al. Single cells make big data: New challenges and opportunities in transcriptomics, *Current Opinion in Systems Biology*, 4, 85-91 <https://doi.org/10.1016/j.coisb.2017.07.004> (2017)

Accuracy is much more important, but in this respect, performance appear to be similar to Seurat?

In terms of accuracy, we demonstrated that SciBet out-performed scmap and Seurat in three aspects:

- i> **Cross-validation for each dataset** (Fig. 2c-d). SciBet achieved much better performance than other two methods in the intuitive comparison, regardless of different performance metrics.
- ii> **Cross-platform.** We benchmarked our algorithms on the human pancreatic datasets for their being profiled by multiple single-cell sequencing platform and also their well-characterized phenotypical knowledge. Our algorithm outperforms the others but not that significantly probably due to the highly-differentiated nature of pancreatic cells, which is not necessarily common in other tissues. In the previous Fig. 2C, we evaluated how these classifiers behave in cross-platform experiments and showed both the classification accuracy and speed, which may leave readers the impression that SciBet out-performs others by orders of magnitude in speed but only shows a slight edge over Seurat v3 in accuracy. For a more intuitive comparison and visualization, we have now represented the cross-platform accuracy in Fig. 3a for each train-test pair separately.
- iii> **False positive control.** Considering that the collection of reference scRNA-seq data may be incomplete, we evaluated whether the classifiers could achieve high prediction accuracy while correctly exclude cells with the type not represented in the training set previously. As shown in Fig. 3f-g, SciBet showed notable superiority by balancing high classification accuracy with a low false positive rate. Such performance is anticipated to significantly improve the quality and rigor of supervised cell type identification.

Reviewer #2 (Remarks to the Author):

This manuscript presents SciBet, a model based supervised cell type identifier. SciBet trains a generative model, which contains an array of probabilities of drawing mRNAs from specific genes. SciBet uses multinomial distributions to model the generation of mRNAs. It assumes that single cell sequencing is a sampling with replacement process: mRNA is sampled from genes with gene-specific probabilities. In training, SciBet computes a set of gene probability profiles, one for each cell type. In testing, given a cell's mRNA profile, SciBet assigns the cell a type by computing a series of posterior probabilities. SciBet iterates through all trained cell types. For each cell type it computes the posterior probability of observing the mRNA profile of the test cell, assuming that the test cell is of the trained cell type. The cell type that generates the highest posterior probability is designated as the type of the test cell. The authors compared against scmap-cluster and Seurat v3 and showed that SciBet has a similar accuracy with Seurat v3 while being significantly faster. It also compared E-test against F-test, M3Drop and showed that E-test improves the accuracy of SciBet when selecting the top 1000-or-fewer genes. This tool is urgently needed in the single cell field because classic clustering algorithm does not annotate the cells into biological category. The web portal is user friendly. The idea of setting a null cell type is interesting. The efficiency of SciBet is impressive (~100x faster than Seurat v3).

We thank the reviewer for the enthusiastic comments on the advantages of SciBet.

I also have some concerns and comments below.

1. The paper did not mention how the general cell population is designed. Do all cell types have equal proportions in the general population? How are under-represented cell types handled?

We collected datasets from the source of the publications with the original cell type annotation, the detailed information has been now listed in Supplementary Tables 1-6. Because the cells were annotated by the unsupervised workflow, which usually consists of clustering, differential expression and cell type identification based on marker gene of each cluster, cell numbers of different cell types rarely have equal proportions. Such unequal proportions reflect true proportions of cells within each dataset, and thus have different contributions to the global assessment of accuracy. To handle this problem, we applied two metrics for the classification performance, the *accuracy score*¹¹ and the *balanced accuracy score*¹¹, where we consider or neglect the contribution of the proportion, respectively. Details can be seen in "Performance assessment by cross-validation" part of the section RESULTS. In our results (Fig. 2c-d), SciBet achieved the best performance regardless the choice of the metric.

In addition, we have selected two well-characterized datasets as case studies (one with

even proportions and another with relatively imbalanced proportions), and benchmarks the confusion matrix of the classification result, in order to demonstrate the advantages of SciBet against scmap and Seurat v3.

Author states “We further collected 42 published human scRNA-seq datasets with full length mRNA coverage and built an integrated dataset to include major human cell types that could be separated clearly by self-projection with SciBet”. There is no information how these 42 reference datasets in the paper.

These data are our best collection of scRNA-seq datasets at the time of project. We have now added the details on data source, publication, sequencing platform and cell number, for the 42 datasets in the Supplementary Table 3.

2. The E-test only shows slight improvements when the number of selected genes for training is small. When the number of genes is beyond 500, E-test have similar, and sometimes even inferior, performance than F-test. Judging from the accuracy plot (Figure 1b), SciBet achieves the highest accuracy with around 2k genes. However, with 2k genes, the performance of E-test shows little difference from F-test. It is not clear why E-test is preferred over F-test.

We thank the reviewer for this helpful comment. It is a major advantage of E-test to outperform other feature-selection methods at the small number of informative genes while keeping high accuracy, and thus to increase computational efficiency for the downstream classification (detailed discussion on the motivation of feature select can be seen in answer 4). The past decade has witnessed an exponential increase in the size of datasets of single-cell sequencing from hundreds to millions, as we mentioned in the first paragraph in Section INTRODUCTION. Selecting informative features is key to minimizing the time cost for the training process and reducing the size of trained model with no performance loss. This will be useful for the ultra-big datasets in the future.

3. It is unclear how this web will be useful for user’s customized datasets that are not presented in the reference datasets. It will be interesting to introduce how to train another model on own data in the tutorial.

Based on the advice from this reviewer, we have now added detailed description in the R documentation for the “SciBet” fuction, which can be used for a one-click function for both training and test with any given dataset.

Classify cells of a given query dataset using a reference dataset.

Description

SciBet main function. Train SciBet with the reference dataset to assign cell types for the query dataset.

Usage

```
SciBet(train, test, k=1000, result=c("list", "table"))
```

Arguments

train The reference dataset, with rows being cells, and columns being genes. The last column should be "label".

test The query dataset. Rows should be cells and columns should be genes.

k Number of genes to select according to the total entropy differences among cell types.

Examples

```
SciBet(train.matr, query.matr)
```

In addition, we have added the online version of this function in our new website (hyperlink "online classification"). Now users can upload their own data for training as well as use our pre-trained models.

Step 2: Choose a reference file

Use pre-calculated reference

Please scroll down and choose a scibet reference by simply clicking the 'choose as reference' button. You can also use the search function of the browser (ctrl+F) for key words to directly locate to your reference of interest. If you are performing cross species analysis, please ensure gene names are compatible with the reference species.

Or Use your own reference

Format of reference file goes the same as test files

选择文件 未选择任何文件
 scibet_mouse_bladder_reference.csv

Reference: Single-cell transcriptomics of 20 mouse organs creates a Tabula Muris

4. The authors introduced E-test without providing a motivation. We suggest the authors to introduce SciBet and its generative multinomial-distribution based mRNA sampling model first. Then give a motivation of why feature selection is important, before introducing E-test. We also prefer the authors to support the motivation of E-test in the Result section, perhaps presenting the prediction accuracy of training SciBet without feature selection.

Thanks for your advice. Feature selection improves relative strength of the biologically meaningful signals against the noisy signals¹², as well as accelerate the process of the downstream analysis such as supervised cell type annotation¹³, that is, the ambient or noisy features influenced by batch effects or other technical variability of may lead to unexpected error for the prediction. Following your advice, we have now modified our logical flow of the revised manuscript by introducing SciBet first in the INTRODUCTION section and then giving the motivation of feature selection in the RESULTS section:

“Because not all genes were equally useful for such the classification problem^{12,13}, we developed E-test to select the cell type-specific genes from the training set in a supervised and parametric manner, in order to remove the noisy genes as well as to accelerate the downstream classification by compressing the model.”

6. The mechanism of E-test is not clearly stated in the main text. We infer that E-test is trying to finding cell-type specific genes that differs from the total population. However, this idea was not clearly stated in the paper.

The reviewer is correct for such inference. We have now realized that in our previous manuscript we failed to explicitly explain the idea behind E-test and refined the description for E-test in the main text as following (in “Overview of the algorithm” part in section RESULTS.):

“... we developed E-test to select the cell type-specific genes from the training set in a supervised and parametric manner... We then proposed a new statistic ΔS , the total entropy difference to measure the deviation of the observed mean expression of cell groups against the null hypothesis, where all cell types were assumed not to be distinct and thus had the same mean and entropy.”

In addition, we have now used the Fig.1b for an intuitive visualization and added detailed description on how the algorithm works and how the benchmarks are performed in “Supervised feature selection by E-test” part in section METHODS.

7. The current manuscript is a bit hard to follow. There are many typos, repeats and undefined concepts. We strongly recommend the authors to significantly improve the writing quality of the manuscript in their revision. In addition, more detailed information (e.g. number of cells, sequencing depth) about reference and test datasets need to be provided.

Since the previous manuscript was directly transferred from another journal, which required a shorter format, the manuscript was constrained by a different word limit and style. We have now taken steps to improve the manuscript quality from the two aspects:

- i) We have refined the structure and modified the logical flow of our manuscript by clarifying the motivation of SciBet first in the INTRODUCTION section, and move the results to the newly-add RESULTS section including 4 parts with subtitles. In the METHODS section, we have specified and clarified the details, definitions, and assumptions to avoid ambiguous or misleading sentences.
- ii) For more detailed information about the datasets used, we added detailed information in the Supplementary Tables, including the data source, publication citation, species & tissue information, sequencing technique and cell number. All codes for generating the benchmarking results has been released at Github: <https://github.com/PaulingLiu/scibet/tree/master/scripts>. For more detailed derivation and discussion, we have added a new section Supplementary Note.

Minor comments:

9. The authors use the term “projection” as cell type annotation. We suggest the authors to avoid this term, as it infers that SciBet is projecting a testing dataset to a training dataset, which is not what SciBet does.

We appreciate this helpful advice and have corrected this misleading word by “classification” or “supervised annotation” in the revised version of manuscript.

10. Line 88-96 is a repeat of the first 3 paragraphs.

We apologize for the repeat and have removed it. In addition, we have now refined the structure of the revised manuscript and carefully examined the problem like such repeats.

11. It is not clear in the main text that SciBet needs to be trained with a curated dataset first. It is also not clear in the main text what is a null dataset and what is its purpose.

As a supervised classifier, SciBet needs a training set (including data and cell label) as the reference first. User can use their own reference dataset or use our pre-trained models. The robustness of SciBet allows the usage of a very large collection of pre-trained models provided on our website. In the revised manuscript, we have clarified this issue in both part “Overview of the algorithm” (Fig. 1) for the workflow and part “Web-based implementation of SciBet” for describing our pre-trained models.

For the second question, we have now added more detailed description for the null datasets in the third paragraph in part “Real-world applications of SciBet” part of section RESULTS as following:

“Due to the incomplete nature of reference scRNA-seq data collection, cell types excluded from the reference dataset may be falsely predicted to be a known cell type. Here we applied a null dataset as background, which is generated by mixing together all cell types in the datasets listed in Supplementary Table 4. For each cell in the test set, we quantified the likelihood to the reference set against that to the null set. Cells with smaller classification confidence score would be assigned as “unassigned cells” and thus be excluded from the downstream classification (METHODS).”

12. Line 75: delta S is not defined.

We have now modified the description in the first paragraph of the RESULTS as following:
“We then proposed a new statistic ΔS , the total entropy difference, to measure the deviation of the observed mean expression of cell groups against the mean expression under the null hypothesis, where all cell types were assumed not to be distinct and thus had the same mean and entropy.”

Detailed definition and description can be seen in “Supervised feature selection by E-test” part in ONLINE METHODS

13. Line 147-148: Sentence “We projected a recent human liver cell 10x genomics dataset to the integrated data by SciBet.” is unclear. Perhaps SciBet trained a model with an integrated data and the model was tested using a 10x liver dataset?

The usage of “project” was indeed misleading. We have now replaced the misleading sentence with the following:

“We annotated the cell type for a recent human liver cell 10x genomics dataset with the integrated data as reference.”

In addition, we have replaced all “project” with “supervised annotation” or “cell type prediction” in the revised manuscript.

14. Line 154-156: Sentence “Furthermore, each major cell type could be further classified into the minor label based on its corresponding dataset.” is not clear. What are major labels and minor labels? How are these labels curated? Are minor labels included in the training dataset or SciBet infers minor labels in an unsupervised manner?

We apologize for the ambiguity. The major cell types were meant to refer to the canonical and well-characterized cell types previously identified without single cell RNA sequencing data and can be usually mapped to the cell type knowledgebase (e.g., EBI Cell Ontology), while the minor cell-types were meant to refer to those novel subtypes uncovered by scRNA-seq with unsupervised clustering and annotation by the original publication. We curated the cell label by unifying the major cell type annotated by different publications (e.g., considering both “T lymphoid cells” and “T cells” as “T cells”) and neglect the minor cell types (e.g., considering both “DC_cluster1” and “DC_cluster2” as DC cells). Realizing the ambiguity of these terms, we have now removed such terminologies and instead used more obvious descriptions for such differences. In the main text, we now added the curation steps above in “Mock human cell atlas” part in section METHODS and replaced the previous statement in the main text with the following one:

“Furthermore, each cell type could be further classified into more precise labels based on datasets uncovering novel sub cell types.”

15. Line 32: unnecessary line break.

We have now removed this unnecessary line break.

16. Some names of functions in this tool are ambiguous, like `C_heatmap()` and `N_heatmap()`.

We have now replaced these ambiguous names with more specific names like **`confusion_heatmap()`** and **`confusion_heatmap_negctrl()`**, and expanded corresponding documentations in the R package.

17. The R documentations of many functions are over-simplified.

We have now expanded the documentation both in the R package and in the website to provide detailed description to all functions used. Here we present an example of the R documentation of `SciBet` function.

SciBet (scibet)

R Documentation

Classify cells of a given query dataset using a reference dataset.

Description

SciBet main function. Train SciBet with the reference dataset to assign cell types for the query dataset.

Usage

```
SciBet(train, test, k=1000, result=c("list", "table"))
```

Arguments

train The reference dataset, with rows being cells, and columns being genes. The last column should be "label".

test The query dataset. Rows should be cells and columns should be genes.

k Number of genes to select according to the total entropy differences among cell types.

Examples

```
SciBet(train.matr, query.matr)
```

18. There are still some little mistakes in the tutorial, like the process of loading model.

We have modified the tutorial following your advice, such as replacing the .rds file of pre-trained "30_major_human_cell_types" with the correct file name, which was used in the documentation of loading model. We thank the reviewer for this helpful comment.

References

- 1 Wagner, G. P., Kin, K. & Lynch, V. J. Measurement of mRNA abundance using RNA-seq data: RPKM measure is inconsistent among samples. *Theory in Biosciences* **131**, 281-285, doi:10.1007/s12064-012-0162-3 (2012).
- 2 Wolf, F. A., Angerer, P. & Theis, F. J. SCANPY: large-scale single-cell gene expression data analysis. *Genome Biol* **19**, 15, doi:10.1186/s13059-017-1382-0 (2018).
- 3 Stuart, T. *et al.* Comprehensive Integration of Single-Cell Data. *Cell* **177**, 1888-1902 e1821, doi:10.1016/j.cell.2019.05.031 (2019).

- 4 Awad, A. *The Shannon entropy of generalized gamma and of related distribution*. (1991).
- 5 Choi, S. C. & Wette, R. Maximum Likelihood Estimation of the Parameters of the Gamma Distribution and Their Bias. *Technometrics* **11**, 683-690, doi:10.1080/00401706.1969.10490731 (1969).
- 6 Huang, M. *et al.* SAVER: gene expression recovery for single-cell RNA sequencing. *Nat Methods* **15**, 539-542, doi:10.1038/s41592-018-0033-z (2018).
- 7 Angerer, P. *et al.* Single cells make big data: New challenges and opportunities in transcriptomics. *Current Opinion in Systems Biology* **4**, 85-91, doi:<https://doi.org/10.1016/j.coisb.2017.07.004> (2017).
- 8 Tabula Muris, C. *et al.* Single-cell transcriptomics of 20 mouse organs creates a Tabula Muris. *Nature* **562**, 367-372, doi:10.1038/s41586-018-0590-4 (2018).
- 9 Cao, J. *et al.* The single-cell transcriptional landscape of mammalian organogenesis. *Nature* **566**, 496-502, doi:10.1038/s41586-019-0969-x (2019).
- 10 Regev, A. *et al.* The Human Cell Atlas. *Elife* **6**, doi:10.7554/eLife.27041 (2017).
- 11 Pedregosa, F. *et al.* Scikit-learn: Machine Learning in Python. *J Mach Learn Res* **12**, 2825-2830 (2011).
- 12 Andrews, T. S. & Hemberg, M. M3Drop: Dropout-based feature selection for scRNASeq. *Bioinformatics*, doi:10.1093/bioinformatics/bty1044 (2018).
- 13 Kiselev, V. Y., Yiu, A. & Hemberg, M. scmap: projection of single-cell RNA-seq data across data sets. *Nat Methods* **15**, 359-362, doi:10.1038/nmeth.4644 (2018).

REVIEWERS' COMMENTS:

Reviewer #1 (Remarks to the Author):

Concerns that were previous raised have been addressed fully by the authors. I have no further comments.

Reviewer #2 (Remarks to the Author):

In general, we are satisfied by the response and the revision. Below are some minor comments regarding the writing of the manuscript. I would suggest authors to proofread the manuscript carefully to minimize typos.

* Line 75-78: Long running sentence. Fix please.

Line 88: multinormal -> multinomial.

Line 92-94: Confusing. Maybe change to "SciBet selects the cell type whose model achieves the highest likelihood/prediction power in describing the distribution of the RNA profile."

Line 124: remove "Then"

Line 139-143: Too long and hard to follow. Change it to "We tested the performance of SciBet on two datasets, with both even and uneven cell type distributions."

Line 155: Change to "We benchmarked SciBet for cross-dataset annotation."

Fix several exaggerations. Such as "showed no sacrifice", "extremely efficient", etc.

REVIEWERS' COMMENTS:

Reviewer #1 (Remarks to the Author):

Concerns that were previous raised have been addressed fully by the authors. I have no further comments.

We thank Reviewer #1 for helping us improve the quality of this paper.

Reviewer #2 (Remarks to the Author):

In general, we are satisfied by the response and the revision. Below are some minor comments regarding the writing of the manuscript. I would suggest authors to proofread the manuscript carefully to minimize typos.

We thank Reviewer #2 for the helpful advice. We have now fixed the following problems and carefully proofread the manuscript.

* Line 75-78: Long running sentence. Fix please.

We have now split this sentence into two separate sentences as following:

“We proposed the null hypothesis where all cell types were assumed not to be distinct and thus had the same mean and entropy. We then proposed a statistic ΔS as the total entropy difference, to measure the deviation of the observed mean expression against the mean expression under the null hypothesis.”

Line 88: multinormal -> multinomial.

We appologize for this mistake and have now fixed it.

Line 92-94: Confusing. Maybe change to “SciBet selects the cell type whose model achieves the highest likelihood/prediction power in describing the distribution of the RNA profile.”

We appreciate the advice and have now updated this sentence as mentioned.

Line 124: remove “Then”

We appologize for this mistake and have now fixed it.

Line 139-143: Too long and hard to follow. Change it to “We tested the performance of SciBet on two datasets, with both even and uneven cell type distributions.”

We have now modified this sentence following the advice raised by Reviewer #2.

Line 155: Change to “We benchmarked SciBet for cross-dataset annotation.”

Thanks for the advice. We have now modified this sentence as mentioned in the revised manuscript.

Fix several exaggerations. Such as “showed no sacrifice”, “extremely efficient”, etc.

We have modified these exaggerations into “showed relatively equivalent performance” and “efficient”, respectively. We have also checked the manuscript carefully to avoid these words.